# Hormetic Effects of Carbendazim on Mycelial Growth and Aggressiveness of *Magnaporthe oryzae*

**DOI:** 10.3390/jof8101008

**Published:** 2022-09-26

**Authors:** Jiehui Song, Chenxi Han, Sijie Zhang, Yan Wang, You Liang, Qigen Dai, Zhongyang Huo, Ke Xu

**Affiliations:** Jiangsu Key Laboratory of Crop Genetics and Physiology & Co-Innovation Center for Modern Production Technology of Grain Crops, Agricultural College, Yangzhou University, Yangzhou 225009, China

**Keywords:** hormetic effect, *Magnaporthe oryzae*, carbendazim, rice, disease management

## Abstract

Rice blast caused by *Magnaporthe oryzae* is one of the most destructive fungal diseases of rice worldwide. Stimulatory effects of low doses of fungicides on pathogens are closely relevant to disease management. In the present study, in potato dextrose agar (PDA) amended with carbendazim at a dose range from 0.003 to 0.3 μg/mL, stimulatory effects on the mycelial growth of three isolates sensitive to carbendazim were tested. Carbendazim at concentrations from 0.003 to 0.1 µg/mL showed stimulatory effects on mycelial growth of isolates Guy11 and H08-1a, while carbendazim at concentrations from 0.003 to 0.03 µg/mL stimulated the growth of isolate P131. The maximum stimulation magnitudes were 11.84% for the three isolates tested. Mycelial colonies grown on PDA amended with different concentrations of carbendazim were incubated at 28 °C in darkness for 7 days as the pretreatment. Pretreatment mycelia were inoculated on fresh fungicide-free PDA and subsequent mycelia growth stimulations were still observed, and the maximum stimulation magnitudes were 9.15% for the three isolates tested. Pretreatment mycelia did not significantly change the tolerance to H_2_O_2_ and NaCl, except that the tolerance to H_2_O_2_ was increased significantly (*p* < 0.05) when the carbendazim was at 0.3 µg/mL. After five generations of mycelial transference on fungicide-free PDA, the transgenerational hormesis of mycelial were exhibited when transferred onto PDA supplemented with carbendazim at 0.3 µg/mL, and the maximum percent stimulation was 51.28%. The time course of infection indicated that the visible initial necrotic symptoms could be detected at 2 DPI on leaves treated with carbendazim at 0.03 µg/mL, whereas no necrotic symptom could be discerned for the control. Statistical results of lesion area and lesion type at 7 DPI showed that there was a significant stimulation (*p* < 0.05) on aggressiveness of *M. oryzae* isolate Guy11 on detached rice leaves at 0.03 µg/mL carbendazim. These results will advance our understanding of hormetic effects of fungicides and provide valuable information for judicious application of fungicides.

## 1. Introduction

Rice is one of the most important food crops in the world and feeds over 50% of the population. However, rice yield is threatened by many plant diseases. In particular, rice blast, which is caused by *Magnaporthe oryzae* (anamorph *Pyricularia oryzae*), is one of the most destructive fungal diseases of rice in the world [1], resulting in at least half of the yield lost in severely infected cases [2]. Rice plants can be infected by *M. oryzae* at any growth stage, causing blast on many parts of the rice, such as the leaf, node, neck, and panicle [3]. Although breeding resistant rice varieties is an economical and efficacious method to control rice blast, the high genetic diversity and quick evolution of *M. oryzae* overcomes rice resistance, causing resistant rice varieties to have a short life [4]. Consequently, the application of fungicides has become an important measure to control rice blast.

Rice blast can be well controlled by many fungicides with different modes of action, such as organophosphorus [5], melanin biosynthesis inhibitors (MBIs) [6], methyl benzimidazole carbamate (MBC) [7], 14α-demethylation inhibitors (DMIs) [8], and quinone outside inhibitors (Qols) [9]. In the early 1970s, the MBC fungicide carbendazim was applied to control rice blast in China. In addition, carbendazim was also frequently used for the control of other rice disease, such as rice sheath blight and rice bakanae disease. Thus, farmers may have applied 3–4 sprays of carbendazim during the whole rice growth season. Extensive and repetitive applications of this fungicide led to the emergence of fungicide resistance in pathogens, causing a reduction or loss of control efficacy of fungicide. For example, resistance to carbendazim was observed in *M. oryzae* field isolates [7]. Although resistance has developed, carbendazim is still used in China, mainly as mixtures with other fungicides of different modes of action, to control rice blast (China Pesticide Information Network, http://www.chinapesticide.org.cn accessed on 14 August 2022).

Hormesis is a phenomenon of stimulation at lower dose and inhibition at higher dose, which reflects a common biological concept. A hormetic model is a biphasic dose response, usually represented by either a J-shaped or an inverted U-shaped dose-response curve [10]. Under natural conditions, exposure of pathogens to low doses of fungicides is almost inevitable. Low-doses of a fungicide may stimulate rather than inhibit mycelial growth and virulence of pathogens [11]. Hormetic-biphasic dose responses have been induced by various fungicides, such as boscalid, carbendazim, dimethachlone, flusilazole, mefenoxam and prochloraz [12,13,14,15,16]. Such hormetic dose responses were found in plant pathogens of profound agricultural importance, e.g., *Botrytis cinerea* [12], *Fusarium graminearum* [17], *Globisporangium ultimum*, *Globisporangium irregular* [15], *Penicillium expansum* [18], *Phytophthora infestans* [19], *Pythium aphanidermatum* [11], and *Sclerotinia sclerotiorum* [14,20]. Low-dose fungicides stimulated mycelia growth, conidia germination rates, sclerotia production, and conidia and mycelia virulence, with the stimulation ranging from below 10% to more than 80%, compared with the control [21]. Hormetic responses were observed for both sensitive and resistant isolates, while the dose range inducing stimulatory responses varied among fungicides. Compared with sensitive isolates, the stimulation amplitude for isolates resistant to dimethachlone was much greater, and the dose range with stimulatory effects was higher in resistant isolates as well [22]. These suggest increased agricultural risks of currently not well understood hormetic responses of phytopathogens to fungicides.

In tests of sensitivity of *M. oryzae* to carbendazim in vitro, we found that carbendazim, at low doses, had significant stimulatory effects on mycelial growth. To date, to our knowledge, there has been no report on stimulations of low doses of carbendazim on mycelial growth or aggressiveness of *M. oryzae*. The objectives of this study were to: (1) determine in vitro stimulatory effects and transgenerational hormesis of low doses of carbendazim on mycelial growth of *M. oryzae*, (2) assess stimulatory effects of low doses of carbendazim on aggressiveness of *M. oryzae*, and (3) explore possible mechanisms for hormetic effects by measuring tolerance to H_2_O_2_ and NaCl of *M. oryzae* exposed to low doses of carbendazim.

## 2. Materials and Methods

### 2.1. Isolates of M. oryzae and Fungicides

Three wild isolates of *M. oryzae* named Guy11, H08-1a and P131 were used in the present study. Technical grade carbendazim (97% active ingredient (a.i.); Tian Jin Jinbei Chemical Co., Ltd. Tianjin, China) was dissolved in 0.1 mol/L HCl to produce a 10,000 μg/mL stock solution. The stock solutions were stored at 4 °C in darkness and prepared within two weeks before experiments.

### 2.2. Rice Varieties and Growth Conditions

The rice, *Oryza sativa* cv. XiangLiangYou 900, susceptible to *M. oryzae* isolate Guy11, was used in aggressiveness assays. The rice seeds were surface sterilized with sodium hypochlorite, washed in distilled water three times and germinated on filter paper in a wet dish under 30 °C for 2 days, then sown into black soil mixed with vermiculite and Pindstrup substrate. The plants grew in a greenhouse with a 12 h photoperiod, 70% relative humidity, and 28 °C/24 °C for day/night, respectively. For the spraying inoculation method, fifth-leaf-stage rice seedlings were used for the fungus inoculation experiments.

### 2.3. Culture of M. oryzae and Conidia Production

All *M. oryzae* isolates were cultured for vegetative growth on potato dextrose agar (PDA) within 2 weeks in complete darkness at 28 °C. For conidia production, the method used was described by Cai et al. [23]. The *M. oryzae* mycelial plugs cut from the PDA plate were cultured on tomato oat agar (TOA) media at 28 °C for 7 days in darkness. Then 1–2 mL of sterilized water was added to the plates and a sterilized coating rod was used to gently scrape the colony. An amount of 500 μL of the mycelium suspension was collected and spread evenly onto fresh TOA medium, and the Petri dishes were incubated at 28 °C for 36 h in darkness. The growing mycelia of *M. oryzae* were scraped off with 1 mL of sterilized deionized water and washed twice with sterilized deionized water. The plates were air dried in the clean bench, covered with a double layer of sterilized gauze, and incubated at 28 °C in light for 48 h. To harvest the conidia, 5 mL sterile deionized water was added and the conidia was brushed down with a sterilized brush. The collection was repeated twice per plate. The conidia suspension was loaded into 10 mL tubes and briefly centrifuged for 30 s. The number of conidia were counted with a hemocytometer.

The PDA was prepared with 200 g of potato, 20 g of dextrose, and 15 g of agar per liter of water. The TOA was prepared with 150 mL of tomato juice, 40 g of oat, and 16 g of agar, used for the production of conidia of *M. oryzae*. 

### 2.4. Sensitivity Detection of M. oryzae to Carbendazim

To determine the sensitivity to carbendazim, *M. oryzae* isolates Guy11, H08-1a and P131 were used to determine the EC_50_ values (the concentration of fungicide causing a 50% reduction in growth, compared with an unamended control) according to the methods described previously [23]. Autoclaved PDA was amended with carbendazim to obtain final concentrations of 0, 0.03, 0.1, 0.3, 1.0, 3.0 and 10.0 μg a.i./mL. An inverted 5 mm diameter mycelial plug, which was cut from the margin of a 7-day-old colony with a cork borer, was inoculated on the PDA amended with carbendazim at the above concentration. The plates were incubated at 28 °C for 7 days in the dark. Each isolate was tested for 4 replicates and the experiment was repeated three times independently. Mean colony diameter (minus 5 mm of the plug diameter) was measured and exhibited as percent growth inhibition. The treatment adding an equal amount of solvent without fungicide was used as control. Percent growth inhibition was expressed by the following formula: (1 − diameter of the treatment/diameter of the control) × 100%. Logarithms of carbendazim concentrations and their corresponding percent growth inhibitions were used to calculate the EC_50_ values by linear regression analysis. 

### 2.5. Stimulatory Effects of Low Doses of Carbendazim on Mycelial Growth of M. oryzae

The 5 mm-diameter mycelial plugs removed from edge of the 7-day-old colonies were inoculated on PDA amended with carbendazim ranging from 0.003–0.3 μg/mL. Carbendazim concentrations were determined according to the results of preliminary experiments, ensuring that at least one concentration could stimulate the radial growth of colony on PDA. Fungicide-free PDA amended with solvent was used as the control. After incubation at 28 °C for 7 days, the diameter of mycelial growth was measured in two perpendicular directions. Inhibitory or stimulatory effects relative to the control were calculated. There were at least 4 replicates for each treatment and the experiment was repeated three times independently.

### 2.6. Effects of Carbendazim Pretreatment on Tolerance of M. oryzae to H_2_O_2_ and NaCl

Determination of sensitivity to H_2_O_2_ and NaCl were assayed according to the previous studies [22,24]. Inverted 5 mm mycelial plugs of isolates H08-1a and P131 removed from 7-day-old colonies were inoculated on PDA amended with carbendazim at 0, 0.003, 0.01, 0.03, 0.1, and 0.3 μg a.i./mL. Mycelial colonies grown on PDA amended with different concentrations of carbendazim were incubated at 28 °C in darkness for 7 days as the pretreatment. Inverted mycelial plugs cut from carbendazim pretreatment colonies were transferred to PDA media amended with H_2_O_2_ at 0, 100, and 200 µg/mL, or NaCl at 0, 200, and 400 µg/mL. The diameter of mycelial growth was measured after incubating at 28 °C for 7 days. There were at least 4 replicates for each treatment and the experiment was repeated three times independently.

### 2.7. Effects of Carbendazim Pretreatment on Mycelial Growth of M. oryzae

Inverted 5 mm mycelial plugs removed from 7-day-old carbendazim pretreatment colonies were inoculated on fresh fungicide-free PDA. The treatment without carbendazim pretreatment was used as the control. After being incubated at 28 °C in darkness for 7 days, the radial growth of the second generation mycelial was measured in two perpendicular directions. This process was continued for five generations of mycelial transference, but new plates were always inoculated with the colony of the previous generation. To test transgenerational hormesis, after five generations, 5 mm mycelial plugs removed from 7-day-old colonies of the sixth generation were transferred to PDA amended with carbendazim at 0.3 or 1.0 μg a.i./mL. Mycelial growth was measured in two perpendicular directions after 7 days of incubation at 28 °C in darkness. Inhibitory or stimulatory effects relative to the fungicide-free PDA were calculated. This experiment was conducted three times with at least 4 replicates for each treatment.

### 2.8. Stimulatory Effects of Spraying Low Doses of Carbendazim on Aggressiveness of M. oryzae on Detached Rice Leaves

The third leaf in the rice seedling was collected from the greenhouse. The detached leaves were cut to 8 cm in length, rinsed with sterile water, air dried on a clean bench, and placed on 9 cm Petri dishes with wet filter paper at the bottom to maintain high humidity. Carbendazim solutions were prepared by diluting the carbendazim stock solution with sterilized 0.1% Triton-100 water. Detached leaves were sprayed with carbendazim solutions at concentrations of 0, 0.003, 0.03, 0.3 and 3.0 µg a.i./mL by a hand-held sprayer. After about 1 h of air drying, leaves were inoculated on the surface by spraying conidial suspension. Conidial suspension of Guy11 was used to evaluate the aggressiveness of *M. oryzae*. Conidia harvested from TOA were resuspended to a concentration of 5 × 10^4^ spores/mL in water solution with 0.01% Tween-20. For inoculation of conidia, 1 mL of conidial suspension was sprayed on each detached rice leaf. Then the Petri dishes were placed in the plant incubator in the dark for 24 h, following a 12 h photoperiod at 28 °C, 85% relative humidity for 7 days. Lesion formation was observed and photographed daily, and recorded by photography 7 days after inoculation.

Two evaluation methods, lesion area and lesion types, were used to assess the severity of rice blast. To count the number of each lesion type, the lesion types were divided into 1–5 types based on their severity: Type 0, no visible blast lesion; Type 1, there were pinhead-sized brown specks on leaves; Type 2, small brown spots within 1.5 mm; Type 3, 2–3-mm gray spots with brown margins; Type 4, diseased lesion with elliptical gray spots longer than 3 mm; Type 5, coalesced lesions infecting more than half of the leaf area.

### 2.9. Statistical Analysis

Percent stimulations of mycelial growth were expressed by the following formula. Percent stimulation (%) = [(mycelial diameter of the treated − diameter of mycelial plugs) / (mycelial diameter of the control − diameter of mycelial plugs) − 1] × 100%. Results were represented as the mean values ± standard deviation. One-way analysis of variance (ANOVA) with a least significant difference (LSD) test in SPSS software (version 21.0; IBM SPSS Inc., Chicago, IL, USA) was used to evaluate the significant differences between treatments.

## 3. Results

### 3.1. Sensitivity to Carbendazim

The carbendazim EC_50_ values of the isolates Guy11, H08-1a and P131 were 0.31, 0.38 and 0.37 μg/mL, respectively, and the mycelia growth of the three isolates were completely inhibited on PDA media amended with carbendazim at 1.0 µg/mL (Table 1). These results revealed that isolates Guy11, H08-1a and P131 were sensitive to carbendazim.

### 3.2. Stimulatory Effects of Low Doses of Carbendazim on Mycelial Growth of M. oryzae

All three isolates grew faster on PDA amended with lower doses of carbendazim than the control. Carbendazim at concentrations from 0.003 to 0.1 µg/mL showed stimulatory effects on mycelial growth of isolates Guy11 and H08-1a, while carbendazim at concentrations from 0.003 to 0.03 µg/mL stimulated the growth of isolate P131 (Figure 1). The largest growth increases of the three isolates were from 8.79 to 11.84%, compared with the control. The largest growth increase for isolate Guy11 was 8.79%, obtained with carbendazim at 0.03 µg/mL; for isolate H08-1a was 11.84%, obtained with carbendazim at 0.1 µg/mL; and for isolate P131 was 11.42%, obtained with carbendazim at 0.01 µg/mL.

### 3.3. Effects of Carbendazim on Tolerance of M. oryzae to H_2_O_2_ and NaCl

After growing on carbendazim-amended PDA media at 0.3 µg/mL for 7 days, sensitivity of *M. oryzae* to H_2_O_2_ at 100 or 200 µg/mL decreased slightly but significantly (*p* < 0.05), compared with the fungicide-free control (Figure 2A). Carbendazim in PDA at mycelial growth stimulating doses of 0.003 to 0.1 µg/mL had no significant (*p* > 0.05) effects on tolerance of *M. oryzae* to H_2_O_2._ For the tolerance of *M. oryzae* to NaCl, there was no significant difference (*p* > 0.05) in mycelial growth on PDA amended with NaCl at 200 or 400 mmol/mL among treatments with different concentrations of carbendazim (Figure 2B). These results indicated that growing on carbendazim-amended PDA at mycelial growth stimulating doses could not enhance tolerance of *M. oryzae* to H_2_O_2_ and NaCl stress.

### 3.4. Effects of Carbendazim Pretreatment on Mycelial Growth of M. oryzae

Carbendazim at the relatively low dosage range had stimulatory effects on the first-generation mycelial growth of the *M. oryzae* isolates. After the stimulated mycelia of first-generation mycelia were inoculated on fresh fungicide-free PDA plates and incubated for 7 days, mycelia growth stimulations were still observed, compared with the control without pretreatment (Table 2), indicating that the hormesis can pass down to the next generation of mycelia. The largest mycelial growth stimulations of three isolates were from 5.10 to 9.15% for the second generation. In our experiments, after five generations of mycelial transference on fungicide-free PDA, there were no obvious stimulations or inhibitions for the sixth generation.

In order to test whether carbendazim pretreatment has transgenerational hormesis on offspring mycelial growth, the sixth-generation mycelia were transferred onto PDA amended with carbendazim at 0.3 or 1.0 µg/mL. At 0.3 µg/mL carbendazim, it was observed that the stimulations of mycelia growth increased in a dose-dependent manner, compared with the control without pretreatment (Figure 3). The largest mycelial growth stimulation of isolate Guy11 reached 51.28%, compared with the control without pretreatment. However, all three isolates treatments could not grow at 1.0 µg/mL carbendazim.

### 3.5. Stimulatory Effects of Low Doses of Carbendazim on Aggressiveness of M. oryzae on Detached Rice Leaves

Visible necrotic symptoms could be detected at 2 DPI on leaves sprayed with carbendazim at 0.03 µg/mL, whereas no necrotic symptom could be discerned for the control (Figure 4). From 4 to 6 DPI, disease symptoms progressed faster on rice leaves treated with carbendazim at 0.03 µg/mL. Statistical results of the lesion area at 7 DPI showed that there was a significant stimulation (*p* < 0.05) of aggressiveness of *M. oryzae* isolate Guy11 on detached rice leaves at 0.03 µg/mL carbendazim, whereas 3.0 µg/mL carbendazim treatment could substantially inhibit infectious fungal growth and limit the lesion area (Figure 5A). Furthermore, the lesions were quantified by dividing the lesions into 1–5 types based on the disease severity. It was observed that 0.03 µg/mL carbendazim treatment also increased the lesions in every type (Figure 5B). Interestingly, 0.3 µg/mL carbendazim treatment could significantly inhibit the mycelial growth of *M. oryzae* in PDA, but promoted the aggressiveness of *M. oryzae* on the detached leaf to a certain extent.

## 4. Discussion

Crump proposed the following criteria for evaluating hormesis: strength of evidence, consistency, soundness of data, and biological plausibility [25]. Thus, evaluation of data is very important for testing hormesis. It seemed that fungicide-resistant isolates were more likely to exhibit hormesis than sensitive isolates. Zhou et al. revealed that both the stimulation amplitude of isolates and the dose range of fungicides inducing stimulatory responses were much higher for resistant isolates than that of sensitive ones [22]. Although only sensitive isolates were used in this study, the curves of *M. oryzae* mycelial growth on PDA and aggressiveness on detached leaves in response to various concentrations of carbendazim showed similar typical hormetic curves and had the same biphasic characteristic.

Analysis of the hormesis database by Calabrese and Blain, which contained nearly 9000 dose responses, revealed that hormetic effects tended to be modest [26]. Other studies collecting hormesis induced by fungicides in plant pathogens summarized that the maximum stimulatory response was generally lower than 60% [21]. The present study showed that low doses of carbendazim could stimulate mycelial growth of *M. oryzae* in vitro and significantly increase aggressiveness on detached rice leaves, and the largest stimulations on mycelial growth by low doses of carbendazim ranged from 8.79 to 11.84% across different isolates. These data indicated that the stimulatory effects of low doses of carbendazim on mycelial growth of *M. oryzae* were consistent with hormesis. However, in the present study, the largest stimulatory effect of aggressiveness (percentage of lesion area) on detached leaf by low doses of carbendazim was about seven-fold greater than the control. Possible reasons for this difference with other studies could be different pathogens, disease evaluation methods and time point.

Oxidative burst is one of the earliest and most common immune responses of plant tissue against an invading pathogen, releasing O_2_^−^ and H_2_O_2_ at the point of pathogen invasion [27]. For example, infection of bean and tomato leaves by *botrytis cinerea* resulted in massive accumulation of H_2_O_2_, both in the plant plasma membrane and in the extracellular sheath covering the surface of fungal hyphae [28,29]. The enhanced tolerance of pathogen to H_2_O_2_ is very likely one of the reasons for the increased aggressiveness to host plants [27]. Some studies showed that subtoxic doses of fungicide pretreatment with single fungicide or mixed fungicides could improve tolerance to H_2_O_2_ [16,22], whereas other studies demonstrated that the aggressiveness-stimulatory effect of subtoxic doses of fungicide was not through enhanced tolerance to oxidative stresses [13,20,30,31]. The present study showed that the doses stimulating the mycelial growth of *M. oryzae* did not change the tolerance of *M. oryzae* to H_2_O_2_, but the dose inhibiting the mycelial growth (0.3 μg/mL carbendazim) increased the tolerance to H_2_O_2_. This result may explain why carbendazim at 0.3 μg/mL inhibited mycelial growth on PDA, but promoted aggressiveness of *M. oryzae* on detached leaves.

Pretreatment is a special form of hormesis [32,33]. Pretreatment activates adaptive response mechanisms that prepare organisms for more intense stress, thus, performing better in subsequent harmful exposures. In *S. sclerotiorum*, pretreatment with flusilazole had hormetic effects for later exposure to flusilazole, prochloraz, tebuconazol [14]. The hormetic effects of pretreatment was also found in *Sclerotinia homoeocarpa* causing dollar spot disease in turf; when 28 isolates were cultured in corn meal agar containing 11 different concentrations of thiophanate-methyl, the mycelia growth of most preconditioned thiophanate-methyl isolates was significantly enhanced by 3–20% [34]. In this study, when the stimulated mycelia grown on carbendazim-amended PDA were inoculated on fungicide-free PDA, mycelia growth stimulations were still observed in the second-generation mycelia (Table 2). This phenomenon was also found in *S. sclerotiorum*, where the mycelia growth stimulations were still observed after the stimulated mycelia grown on low doses of boscalid-amended PDA were inoculated on fresh fungicide-free PDA [13]. 

Even after five generations of mycelial transference on fungicide-free PDA, mycelial growth of the sixth generation was stimulated on PDA amended with carbendazim, compared with the control without pretreatment (Figure 3). Interestingly, stimulation of colony growth of Guy11 reached 51.28%, compared with the control. However, this was still within the reasonable range of hormetic effect, according to previous relevant studies [21]. As for possible mechanisms of transgenerational hormesis, organisms may maintain or enhance evolutionary fitness of themselves and their offspring in stressful environments via hormetic mechanisms, mediated via molecular mechanisms, such as changes in gene expression, transposable element activity, genetic recombination, and non-lethal mutations, as well as evolutionary rescue or phenotypic plasticity strategies [35,36,37]. In addition, in nematode *Caenorhabditis elegans*, the acquisition and inheritance of hormetic information was regulated by small RNA-based inter-tissue communication [38]. However, the mechanism of transgenerational hormesis in *M. oryzae* needs further study. Consequently, fungicide pretreatment increased tolerance of pathogen offspring to renewed exposure to fungicides, suggesting that hormesis can also offer transgenerational benefit. Furthermore, the low-dose stimulatory responses can eventually lead to the development of resistance to environmental challenges, which may act as a driver of evolution, thus, the application risks of fungicide need to be further evaluated in the future.

The stimulations on aggressiveness of *M. oryzae* by low doses of fungicides have important impacts on the formulation of plant disease control strategies. In the field, some plant pathogens would inevitably be exposed to low doses of fungicide, such as application drift, dilution of fungicide by rainwater, low-dose applications to reduce costs, inappropriate fungicide applications, and other reasons [11]. The potential effects of fungicide hormesis are very harmful to food security, as they may lead to greater disease incidence and severity, higher levels of mycotoxins, and larger crop losses [17]. In the present study, the aggressiveness of isolate Guy11 on detached leaf was significantly enhanced by low doses of carbendazim. Nevertheless, the current study is insufficient to estimate the prevalence and quantitative characteristics of hormesis in *M. oryzae*, and further studies are needed. Understanding the quantitative and qualitative features of hormesis induced by fungicide, and integrating the hormesis concept into the formulation of plant disease control strategies, will improve the application efficiency of fungicides.

As for the mechanism of hormesis, direct stimulation or an overcompensation to a disruption of homeostasis may be two mechanisms responsible for hormetic dose–response [39]. In a previous study which reported that herbicide phosphinothricin stimulated growth of plant *Lotus corniculatus* L, a mechanism of direct stimulation for chemical hormesis was reported. Lower concentrations of herbicide activated the chloroplast isoform GS2, causing growth stimulation, while higher concentrations inhibited chloroplast isoforms GS1 and GS2, resulting in growth inhibition [40]. In pharmacology literature, a high amount of low-dose stimulatory responses may be relevant to the direct stimulation mechanism [41]. Some previous studies on the aggressiveness stimulations of fungicide hormesis on *S. sclerotiorum* or *B. cinerea* indicated that stimulatory effects of carbendazim [30,42], trifloxystrobin [43], and flusilazole [31] belonged to a direct stimulation mechanism. In general, overcompensation to disruption of homeostasis is manifested as early inhibition before stimulation. It was reported that a chemical stressor inhibited plant growth at 24 h but stimulated growth at 72 h [44]. In this study, the visible necrotic symptoms on rice leaves sprayed with carbendazim at 0.03 µg/mL could be observed as early as 2 DPI. The fact revealed that stimulation of *M. oryzae* by low doses of carbendazim was due to a direct stimulation rather than an overcompensation to disruption of homeostasis.

## Figures and Tables

**Figure 1 jof-08-01008-f001:**
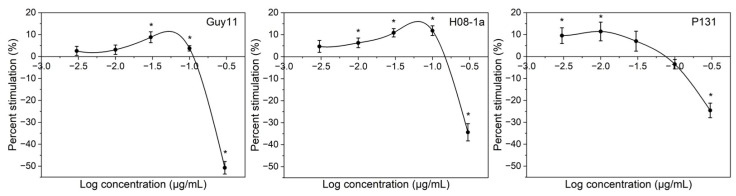
Stimulatory effects of carbendazim in PDA on the mycelial growth of *Magnaporthe oryzae*. Error bars represent the standard errors of mean values in independent experiments. Asterisk indicates significant difference (*p* < 0.05) between the treatment and the control.

**Figure 2 jof-08-01008-f002:**
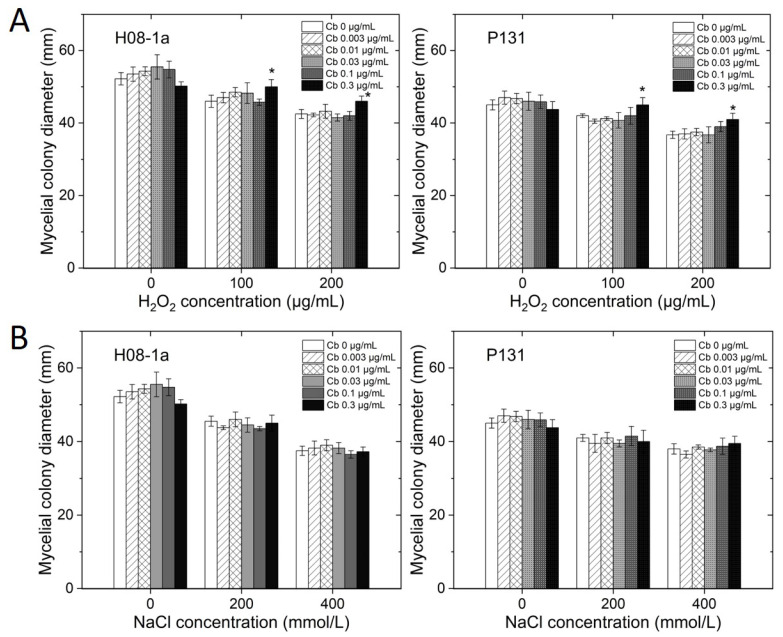
Effects of carbendazim on tolerance of *Magnaporthe oryzae* isolates H08-1a and P131 to H_2_O_2_ (**A**) and NaCl (**B**). Error bars represent the standard errors of mean values in independent experiments. Asterisk indicates significant difference (*p* < 0.05) between the treatment and the control without pretreatment.

**Figure 3 jof-08-01008-f003:**
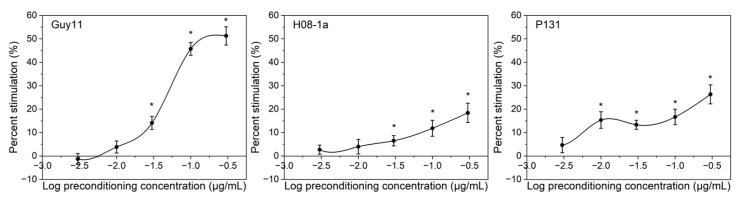
Stimulatory effects of carbendazim pretreatment on the sixth-generation mycelial growth of *Magnaporthe oryzae* grown on PDA supplemented with carbendazim at 0.3 µg/mL. Error bars represent the standard errors of mean values in independent experiments. Asterisk indicates significant difference (*p* < 0.05) between the treatment and the control without pretreatment.

**Figure 4 jof-08-01008-f004:**
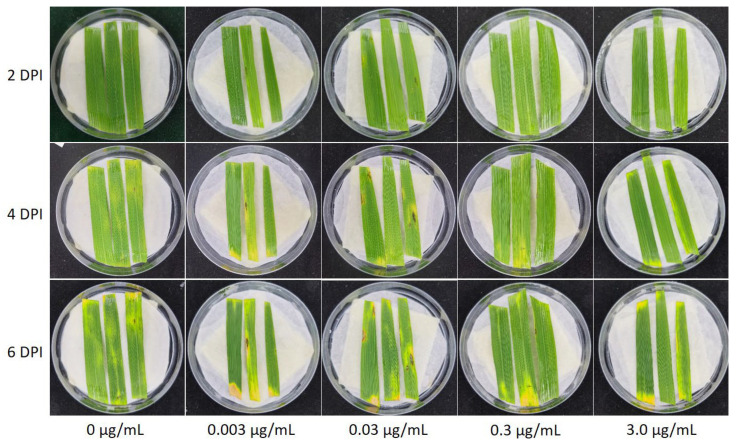
Time course of the infection process of conidia of *Magnaporthe oryzae* isolate Guy11 on detached rice leaves. Leaves were sprayed with carbendazim solutions at concentrations of 0, 0.003, 0.03, 0.3, and 3.0 µg/mL before inoculation of *M. oryzae*. DPI = days post inoculation.

**Figure 5 jof-08-01008-f005:**
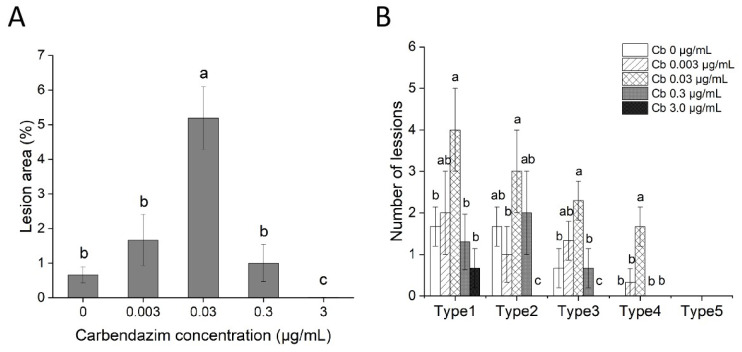
Quantification of lesion area (**A**), and lesion type (**B**), of *Magnaporthe oryzae* isolate Guy11 on detached rice leaves at 7 DPI. Error bars represent standard deviation, and different letters represent significant differences (*p* < 0.05).

**Table 1 jof-08-01008-t001:** Sensitivity of *Magnaporthe oryzae* isolates to carbendazim.

Isolate	Phenotype	Carbendazim EC_50_ (μg/mL) ^y^	MIC (μg/mL) ^z^
Guy11	Sensitive	0.31	1.0
H08-1a	Sensitive	0.38	1.0
P131	Sensitive	0.37	1.0

^y^ EC_50_: effective concentration for 50% inhibition of mycelial growth. ^z^ MIC: minimum inhibitory concentration.

**Table 2 jof-08-01008-t002:** Percent stimulations of carbendazim in PDA for the first- and second-generation mycelial growth.

Isolate	Stimulations (%)				
	Carbendazim Concentration in PDA (μg/mL)				
	0.003	0.01	0.03	0.1	0.3
Guy11					
First generation	2.56 (2.04) ^z^	3.07 (2.15)	8.79 (2.49)	3.73 (1.22)	−51.74 (2.87)
Second generation	1.02 (1.58)	2.65 (1.09)	5.10 (2.12)	2.53 (1.33)	−3.67 (1.08)
H08-1a					
First generation	4.69 (2.73)	6.30 (2.19)	10.91 (1.92)	11.84 (2.19)	−34.38 (3.91)
Second generation	3.38 (1.97)	5.15 (1.22)	9.15 (3.50)	8.42 (2.65)	−2.54 (0.98)
P131					
First generation	9.53 (3.60)	11.42 (4.26)	7.02 (4.56)	−3.51 (2.16)	−24.56 (3.33)
Second generation	4.27 (1.35)	6.10 (2.24)	2.44 (2.11)	2.13 (2.38)	−2.74 (1.17)

^z^ Data in parentheses represent the standard errors of mean values in independent experiments.

## Data Availability

The data presented in this study are included in the article, further inquiries can be directed to the corresponding author.

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
