# Peer review of "Hormetic Effects of Carbendazim on Mycelial Growth and Aggressiveness of Magnaporthe oryzae"

_jof, 2022, doi:10.3390/jof8101008_

Round 1

Reviewer 1 Report

Dear Authors,

The submitted paper entitled "Hormetic Effects of Carbendazim on Mycelial Growth and Virulence of Magnaporthe Oryzae" reported an interesting and important observation and analysis of fungicide carbendazim effects on Magnaporthe Oryzae physiology. The obtained results provide valuable information for the fungicide application based on the hormetic effects of fungicides. Generally, this paper presents new findings with well-supported conclusions, well-organized experiments and smooth English writing. So my suggestion is that the present manuscript can be accepted for publication with several minor revisions (please see below).

1. Please check the two phrases ‘hormetic effects’ (in title) and ‘stimulatory effect’ (in key words list). If there is no difference between them, please make unify through the whole manuscript.

2. What means ‘a.i.’ (line 89, 124)? Please give full-name or necessary explanation.

3. Line 201, the unit of EC50 is missed, i.e., µg/mL, please add it and check through the whole manuscript.

4. Line 221, ‘statistically’ before ‘significantly’ can be deleted.

5. The expression ‘detached rice leaf’ can be changed as ‘detached rice leaves’.

6. The word ‘preconditioning’ can be changed as ‘pretreatment’. And ‘nonpreconditioned’ can be expressed as ‘non-pretreatmented’ or ‘*** without pretreatment’.

Author Response

The submitted paper entitled "Hormetic Effects of Carbendazim on Mycelial Growth and Virulence of Magnaporthe Oryzae" reported an interesting and important observation and analysis of fungicide carbendazim effects on Magnaporthe Oryzae physiology. The obtained results provide valuable information for the fungicide application based on the hormetic effects of fungicides. Generally, this paper presents new findings with well-supported conclusions, well-organized experiments and smooth English writing. So my suggestion is that the present manuscript can be accepted for publication with several minor revisions (please see below).

Respond: Thanks very much for your kindly comments.

  1. Please check the two phrases ‘hormetic effects’ (in title) and ‘stimulatory effect’ (in key words list). If there is no difference between them, please make unify through the whole manuscript.

Respond: Hormetic effect includes stimulation at lower dose and inhibition at higher dose, which is different from stimulatory effect. So we changed “stimulatory effect” to “hormetic effect” in keywords.

  1. What means ‘a.i.’ (line 89, 124)? Please give full-name or necessary explanation.

Respond: Thanks for your suggestion. It means active ingredient, and we added the full-name where it first appeared.

  1. Line 201, the unit of EC50 is missed, i.e., µg/mL, please add it and check through the whole manuscript.

Respond: We revised and checked the whole manuscript.

  1. Line 221, ‘statistically’ before ‘significantly’ can be deleted.
  2. The expression ‘detached rice leaf’ can be changed as ‘detached rice leaves’.
  3. The word ‘preconditioning’ can be changed as ‘pretreatment’. And ‘nonpreconditioned’ can be expressed as ‘non-pretreatmented’ or ‘*** without pretreatment’.

Respond: Revised based on your suggestions.

Reviewer 2 Report

The article focuses on a relevant topic, namely, stimulation of mycelium growth and conidiagenesis of the plant pathogenic fungus Magnaporthe oryzae with low doses of the carbendazim fungicide as well as an increase in its aggressiveness when infecting detached rice leaves. This phenomenon is called the hormetic effect. The relevance of the chosen topic is due to the fact that during industrial treatments, a part of the targeted pathogen population may be exposed by reduced doses of the fungicide because of demolition of the formulation during spaying, washing off with rain and other factors. In this case, there is a risk of reverse action of the fungicide. In the future, the study of the hormesis phenomenon will allow assessment the risks of an undesirable hormesis effect in fungicidal treatments.

Hormosis is indicated for a number of economically important pathogens, including phytopathogenic those, and related contact fungicides, such as including carbendazim when used against a number of phytopathogenic fungi. This article shows for the first time that hormosis also occurs when carbendazim is used against M. oryzaeThere are some comments on the content of this article.

A general comment is related to the term "virulence". This term is used in the scientific literature when referring to the ability of a pathogen to infect certain varieties of a host plant. The intensity of damage by a pathogen of a susceptible variety would be more correctly called  aggressiveness. Thus, this article describes the stimulation of the aggressiveness of pathogens by lowered fungicidal doses.

Comment to Fig.1 - What do the asterisks above some dots mean? An explanation should have been given in the caption of the figure.

Comment to Fig. 2 - Why are the results shown for only two of the three isolates M. oryzae ?

Comment to Fig. 3 - What do the stars above the columns show?

The figure 3 shows the most interesting results of this work. As can be seen from the presented data, the growth of isolate Guy 11 on PDA was significantly stimulated with 0.3 mkg/ml of carbendazim compared to non-treated control after a single inoculation on the medium with carbendazim and five subsequent subcultures on the fungicide-free media. Stimulation of colony growth on a solid medium reached 51.28% compared to the control. It would make sense for the authors to discuss in more detail possible mechanisms of this very interesting phenomenon.

Another question concerns the different behavior of Guy 11 and two other isolates used as presented on Fig. 3. The stimulation of mycelial growth of these two isolates is approximately half that of the Guy 11 isolate. The data may indicate the strain specificity of the hormoses trait, which is unlikely. It would make sense for the authors to discuss these data.

In the section Discussion (lines 326-331) the authors write: “... mycelial growth stimulations were still observed in the second generation mycelia (Fig. 3). The reference to Fig. 3 seems to be wrong and should be replaced by reference to Table 2.

The sentence (lines 329-331) erroneously ends with a reference to Fig. 4, instead of correct reference to Fig. 3.

Author Response

The article focuses on a relevant topic, namely, stimulation of mycelium growth and conidiagenesis of the plant pathogenic fungus Magnaporthe oryzae with low doses of the carbendazim fungicide as well as an increase in its aggressiveness when infecting detached rice leaves. This phenomenon is called the hormetic effect. The relevance of the chosen topic is due to the fact that during industrial treatments, a part of the targeted pathogen population may be exposed by reduced doses of the fungicide because of demolition of the formulation during spaying, washing off with rain and other factors. In this case, there is a risk of reverse action of the fungicide. In the future, the study of the hormesis phenomenon will allow assessment the risks of an undesirable hormesis effect in fungicidal treatments.

Hormosis is indicated for a number of economically important pathogens, including phytopathogenic those, and related contact fungicides, such as including carbendazim when used against a number of phytopathogenic fungi. This article shows for the first time that hormosis also occurs when carbendazim is used against M. oryzae. There are some comments on the content of this article.

  1. A general comment is related to the term "virulence". This term is used in the scientific literature when referring to the ability of a pathogen to infect certain varieties of a host plant. The intensity of damage by a pathogen of a susceptible variety would be more correctly called aggressiveness. Thus, this article describes the stimulation of the aggressiveness of pathogens by lowered fungicidal doses.

Respond: Thanks for your professional advice, we changed “virulence” to “aggressiveness”.

  1. Comment to Fig.1 - What do the asterisks above some dots mean? An explanation should have been given in the caption of the figure.

Respond: Thank you. An explanation added in the caption of the figure.

  1. Comment to Fig. 2 - Why are the results shown for only two of the threeisolates oryzae ?

Respond: First, considering the workload was very large, we only tested two isolates. For each isolate, there were two compounds (H2O2 and NaCl), three treatments, six pretreatment concentrations, with at least four replicates for each treatment and the experiment was repeated three times independently. Next, before the test, we had referred to several literatures (references 13, 20, 22, 31), in which the author selected only one or two isolates as representatives. Finally, the results of the two tested isolates were consistent, so we didn't test more isolates.

  1. Comment to Fig. 3 - What do the stars above the columns show?

Respond: An explanation added in figure 2 and 3.

  1. The figure 3 shows the most interesting results of this work. As can be seen from the presented data, thegrowth of isolate Guy 11 on PDA was significantly stimulated with 0.3 mkg/ml of carbendazim compared to non-treated control after a single inoculation on the medium with carbendazim and five subsequent subcultures on the fungicide-free media. Stimulation of colony growth on a solid medium reached 51.28% compared to the control. It would make sense for the authors to discuss in more detail possible mechanisms of this very interesting phenomenon.

Respond: Yes, we agree with you. It was very interesting that stimulation of colony growth reached 51.28% compared with the control. But this was still within the reasonable range of hormetic effect, according to previous relevant studies. For example, some studies collecting hormesis induced by fungicides in plant pathogens summarized that the maximum stimulatory response was generally lower than 60% (references 21). As for possible mechanisms of transgenerational hormesis, organisms may maintain or enhance evolutionary fitness of themselves and their offspring in stressful environments via hormetic mechanisms, mediated via molecular mechanisms, such as changes in gene expression, transposable element activity, genetic recombination, and nonlethal mutations, as well as evolutionary rescue or phenotypic plasticity strategies (references 35-37). Besides, in nematode Caenorhabditis elegans, the acquisition and inheritance of hormetic information was regulated by small RNA-based intertissue communication (references 38). However, the mechanism of transgenerational hormesis in M. oryzae needs further study. We added these in Discussion (lines 341–352).

  1. Another question concerns the different behavior of Guy 11 and two other isolates used as presented on Fig. 3. The stimulation of mycelial growth of these two isolates is approximately half that of the Guy 11 isolate. The data may indicate the strain specificity of the hormoses trait, which is unlikely. It would make sense for the authors to discuss these data.

Respond: The stimulation of the sixth generation mycelial growth of Guy 11 was much higher than that of the other two isolates. We think this was due to the individual specificity of hormesis trait. Besides, Guy11 was more sensitive to carbendazim than other two isolates. As shown in Fig. 3, the stimulations of mycelia growth increased in a dose-dependent manner. It can be predicted that the other two isolates will obtain higher stimulation effect at higher carbendazim pretreatment concentrations.

  1. In the section Discussion (lines 326-331)the authors write: “... mycelial growth stimulations were still observed in the second generation mycelia (Fig. 3). The reference to Fig. 3 seems to be wrong and should be replaced by reference to Table 2.
  2. The sentence (lines 329-331) erroneously ends with a reference to Fig. 4, instead of correct reference to 3.

Respond: Sorry, we corrected these mistakes.